# Platelet Reactivity and Cardiovascular Mortality Risk in the LURIC Study

**DOI:** 10.3390/jcm12051913

**Published:** 2023-02-28

**Authors:** Martin Berger, Alexander Dressel, Marcus E. Kleber, Winfried März, Peter Hellstern, Nikolaus Marx, Katharina Schütt

**Affiliations:** 1Department of Internal Medicine I, University Hospital Aachen, RWTH Aachen University, 52074 Aachen, Germany; 2D-A-CH-Gesellschaft Prävention von Herz-Kreislauf-Erkrankungen e.V., 22765 Hamburg, Germany; 3Dr. Dressel Consulting, 68167 Mannheim, Germany; 4Vth Department of Medicine (Nephrology, Hypertensiology, Rheumatology, Endocrinology, Diabetology), Medical Faculty Mannheim, University of Heidelberg, 68167 Mannheim, Germany; 5SYNLAB MVZ Humangenetik Mannheim GmbH, 68167 Mannheim, Germany; 6Clinical Institute of Medical and Chemical Laboratory Diagnostics, Medical University Graz, 8036 Graz, Austria; 7Center of Hemostasis and Thrombosis Zurich, 8006 Zurich, Switzerland

**Keywords:** platelet reactivity, CD63, CD62p, Fibrinogen-binding, cardiovascular mortality

## Abstract

Background: The clinical and prognostic implications of platelet reactivity (PR) testing in a P2Y_12_-inhibitor naïve population are poorly understood. Objectives: This explorative study aims to assess the role of PR and explore factors that may modify elevated mortality risk in patients with altered PR. Methods: Platelet ADP-induced CD62P and CD63 expression were measured by flow-cytometry in 1520 patients who were referred for coronary angiography in the Ludwigshafen Risk and Cardiovascular Health Study (LURIC). Results: High- and Low-platelet reactivity to ADP were strong predictors of cardiovascular and all-cause mortality and risk equivalent to the presence of coronary artery disease. (High platelet reactivity 1.4 [95% CI 1.1–1.9]; Low platelet reactivity: 1.4 [95% CI 1.0–2.0]). Relative weight analysis indicated glucose control (HbA1c), renal function ([eGFR]), inflammation (high-sensitive C-reactive protein [hsCRP]) and antiplatelet therapy by Aspirin as consistent mortality risk modifiers in patients with Low- and High-platelet reactivity. Pre-specified stratification of patients by risk modifiers HbA1c (<7.0%), eGFR (>60 mL/min/1.73 m^2^) and CRP (<3 mg/L) was associated with a lower mortality risk, however irrespective of platelet reactivity. Aspirin treatment was associated with reduced mortality in patients with high platelet reactivity only (*p* for interaction: 0.02 for CV-death [<0.01 for all-cause mortality]. Conclusions: Cardiovascular mortality risk in patients with High- and Low platelet reactivity is equivalent to the presence of coronary artery disease. Targeted glucose control, improved kidney function and lower inflammation are associated with reduced mortality risk, however independent of platelet reactivity. In contrast, only in patients with High-platelet reactivity was Aspirin treatment associated with lower mortality.

## 1. Introduction

The clinical and prognostic implications of platelet reactivity testing in a P2Y_12_-inhibitor naïve population are unclear. While clinical and experimental findings suggest that platelet reactivity by P2Y_1_ and P2Y_12_ receptor-dependent purinergic signaling predicts atherothrombosis and cardiovascular mortality, routine platelet reactivity testing is not recommended by the current ACC/AHA guidelines and ESC guidelines [1,2,3,4,5,6]. This recommendation mainly arose from large clinical trials (i.e., TRIGGER-PCI, ARCTIC, ANTARCTIC, GRAVITAS), which demonstrated that escalation of antiplatelet therapy in patients with residual-high P2Y_12_ reactivity failed to demonstrate a significant clinical benefit in PCI settings [7,8,9,10,11]. Interestingly, a prespecified sub-study of the ARCTIC trial demonstrated that treatment intensification halved residual P2Y_12_ reactivity but did not result in a significant cardiovascular benefit [12]. This lack of a clinical benefit suggests other factors potentially confound the relationship between platelet reactivity and cardiovascular event rates. Therefore, we hypothesized that platelet reactivity might constitute a biomarker that is altered by a distinct cardiovascular risk profile. Following this hypothesis, identifying and adjusting risk modifiers may serve as a rational therapeutic strategy to address atherothrombotic and cardiovascular mortality risks.

Therefore, the aim of the current explorative study was to identify and investigate the role of risk modifiers on mortality in patients with abnormal platelet reactivity and assess these for the potential of cardiovascular mortality risk reduction. 

## 2. Methods

An extended description of clinical characteristics and methodology is available in the Appendix A.

### 2.1. Study Design, Participants and Clinical Characterization

A total of 3316 patients, who were referred for coronary angiography to the Ludwigshafen Heart Center in Germany, were recruited between July 1997 and January 2000 [13]. The study was approved by the ethics committee at the Ärztekammer Rheinland-Pfalz and was conducted in accordance with the Declaration of Helsinki. All participants provided written informed consent. Inclusion criteria were: German ancestry, clinical stability except for acute coronary syndromes, and the availability of a coronary angiogram. Individuals suffering from any acute illness other than acute coronary syndromes, chronic non-cardiac diseases, or malignancy within the past five years and those unable to understand the purpose of the study were excluded. Clinically relevant coronary artery disease (CAD) was defined as the occurrence of ≥1 stenosis of ≥50% in ≥1 of 15 coronary segments. Individuals with stenoses < 20% were considered not to have CAD. 

### 2.2. Reagents

Adenosine diphosphate (ADP) and thrombin receptor activating pepetide-6 (TRAP) were from Sekisui Virotech (Ruesselsheim, Germany) and from Bachem (Bubendorf, Switzerland), respectively. The monoclonal antibodies against CD41-PC7 and CD63-PE were purchased from Beckman Coulter (Krefeld, Germany). The antibodies against CD62P-PC5 and Fibrinogen-FITC were from BioLegend (Koblenz, Germany).

### 2.3. Platelet Reactivity Testing

Platelet reactivity was tested on the day coronary angiography was performed, and blood was drawn before the procedure by atraumatic venipuncture using a modification of a procedure described previously [14]. Briefly, after discarding the first 2 mL, citrated whole blood (1 part 0.106 mmol/L sodium citrate + 9 parts blood) was obtained using Sarstedt monovettes (Nümbrecht, Germany). After resting at 37 °C for 30 min, the whole blood was diluted 1:10 with calcium-free Tyrode’s buffer. Immediately thereafter, 36 µL diluted whole blood was mixed with 4 µL of 100 µM ADP or 100 µM thrombin receptor activating pepetide-6 (TRAP) to achieve final concentrations of 10 µM of ADP or TRAP, respectively, and subsequently incubated for 10 min at 37 °C. After the addition of 20 µL of antihuman monoclonal antibodies against CD62P-PC5, CD63-PE, or Fibrinogen-FITC and further incubation for 5 min at 37 °C in the dark, the reaction was quenched with 1 ml of ice-cold Tyrode’s buffer. Platelet activity marker expression was measured on a Coulter Epics XL flow cytometer (Coulter, Krefeld, Germany) and expressed as median fluorescence intensity (MFI) relative to the basal expression of the respective activity marker. Ten-thousand single platelets were gated based on forward and sideward-scatter characteristics and CD41 (PC-7 labeled) expression. Prior to analysis, MFI was normalized to correct for lot-to-lot variations by fluorescent beads. Ten-thousand single platelets were gated based on forward and sideward-scatter characteristics and CD41-PC-7 expression. Isotype controls were used to estimate non-specific binding. To prevent the false-positive classification of ADP-induced platelet reactivity, patients on the P2Y_12_ inhibitor clopidogrel (N = 104, 5.8%) and those who were unresponsive (i.e., preactivated) to ADP-stimulation (defined as ADP-induced activation marker expression relative to basal marker expression < 1.0; N = 156, 8.7%) were excluded from any subsequent analysis (Appendix A). Therefore, platelet reactivity data were available in 1520/1780 (85.4%) patients.

### 2.4. Follow-Up and Endpoints

There was a follow-up for all-cause and cardiovascular mortality. Information on the vital status was obtained from local person registries. Using death certificates, two experienced clinicians independently classified the causes of death. They were blinded to any data of the study participants. In cases of disagreement or uncertainty concerning the coding of a specific cause of death, classification was made by a principal investigator of the LURIC study (W. M.). In 1520 patients, a total of 452 deaths (284 of cardiovascular cause) occurred during a mean follow-up time of 8.4 (±2.8) years.

### 2.5. Statistical Analysis

The baseline characteristics are reported as median with inter-quartile ranges and percentages in cases of categorical variables and means with standard deviations (resp. medians with inter-quartile ranges) in cases of continuous variables. Continuous variables were tested for normal distribution using the Kolmogorov–Smirnov test and log-transformed before analysis when required. Data were stratified into three groups according to platelet reactivity (i.e., High-, Low-platelet reactivity, and Reference group). Platelet CD63 expression demonstrated better discrimination for Low-platelet reactivity than CD62P and was therefore chosen for stratification (cf. results for details). The High- and Low-platelet reactivity groups were derived from an age- and sex-adjusted Cox model (model 1) by restricted cubic splines (RCS) when hazard ratios for cardiovascular mortality significantly deviated from 1.0 in Low (N = 464) and High (N = 511) CD63 expressors (Appendix A). The remaining patients were considered as the Reference group (N = 545). Cut-offs were <1.22 for Low-platelet reactivity, 1.22–1.50 for the Reference group, and >1.50 for High-platelet reactivity (i.e., ADP-CD63 [MFI]/basal CD63 [MFI]). Comparisons between High-, Low-platelet reactivity and the Reference group were made with the Kruskal-Wallis ranks sum test for continuous variables and with the two-sided Fisher test for binary variables (cf. Appendix A for details). The RCS was set up to have five knots at the 5th, 27.5th, 50th, 72.5th, and 95th percentile for CD62P and CD63 expression, respectively. Splines were adjusted for potential confounders as follows: In a basic model (model 1), we adjusted for age and sex. In a second model (model 2), we additionally adjusted to living conditions (alone/not alone), smoking, alcohol intake, and anti-platelet therapy. In a third model, we adjusted for the glomerular filtration rate (eGFR), hsCRP, low-density lipoprotein cholesterol (LDL-C), HbA1c%, body mass index (BMI), and systolic blood pressure. In the last model (model 4), we additionally adjusted for the Charlson comorbidity Index and Friesinger Score. The stability of all models was confirmed by repeated random resampling (cf. Appendix A for details). Relative weight analysis was performed for each of the 3 groups (i.e., High-, Low-platelet reactivity, and Reference group) individually according to the methodology by Heller et al. as previously described [15,16]. To omit multiple testing, only modifiable factors with an R^2^ > 0.01 were used for further assessment. Based on these risk modifiers, risk groups were created. Cut-offs for risk groups were determined a priori based on a guideline recommendation [17] for treatment targets and risk markers (e.g., HbA1c < 7% in patients with diabetes mellitus; eGFR < 60 mL/min) or observational studies [18] (i.e., hsCRP < 3 mg/L). In any case, where Cox regression has been performed, the global test for the proportional hazard assumption by scaled Schönfeld residuals provided a *p*-value larger than 0.05. (cf. Appendix A for details). All analyses were performed using the SPSS 26.0 statistical package (IBM SPSS Inc., Ehningen, Germany) and R (version 4.0.2). All statistical tests were 2-sided, and *p* values <0.05 were considered significant. 

## 3. Results 

### 3.1. Platelet Reactivity and Cardiovascular and All-Cause Mortality in the LURIC Study

Among the 1520 patients investigated, 452 deaths occurred (284 of cardiovascular cause) within a mean follow-up time of 8.4 ± 2.8 years. ADP-induced platelet reactivity at baseline measured by Fibrinogen-binding, CD62P, and CD63 expression was a significant predictor of cardiovascular and all-cause mortality, and all three markers demonstrated a U-shaped relationship with cardiovascular mortality rates (Appendix A). CD63 expression was a better discriminator for patients with Low-platelet reactivity and was thus chosen for all subsequent analyses to assess the full range of abnormal platelet reactivity (Figure 1A,B). Based on the hazard for cardiovascular mortality, patients were stratified into a High- and Low-platelet reactivity risk population (High platelet reactivity: 511 [33.6%]; Low platelet reactivity: N = 464 [30.52%]). Within these risk groups, the association of High- and Low platelet reactivity with cardiovascular mortality remained significant after additional adjustment for age, sex, living conditions, smoking, alcohol, antiplatelet therapy, eGFR, hsCRP, LDL-C, HbA1c, systolic blood pressure, Charlson Comorbidity Index and Friesinger Score (Model 4: Low-platelet reactivity: 1.4 [95% CI 1.0–2.0], *p* = 0.0254 [1.2 for death]; High-platelet reactivity: 1.4 [95% CI 1.1–1.9], *p* = 0.0175 [1.2 for death]) (Figure 1C,D). TRAP-induced platelet reactivity was not predictive of cardiovascular and all-cause mortality (Appendix A).

### 3.2. Baseline Characteristics in Patients with Abnormal Platelet Reactivity

Clinical and laboratory characteristics stratified by ADP-induced platelet reactivity are shown in Table 1. Atherosclerosis-associated diseases and heart failure were equally distributed among all groups at baseline. However, patients with High-platelet reactivity had a distinct cardiovascular risk profile with a significantly higher prevalence of chronic kidney disease (CKD) and diabetes mellitus as well as a higher Framingham Risk Score (FRS) and Charlson Comorbidity Index (CCI). In addition, plasma levels of HbA1c and hsCRP were significantly higher in the High-platelet reactivity group, whilst eGFR was lower. In contrast, clinical characteristics of patients with Low-platelet reactivity were statistically equivalent to the reference group. Patients that received aspirin at baseline were equally distributed among all groups, whilst the proportion of patients that received diuretics was significantly higher in the High-platelet reactivity group (Table 2). 

### 3.3. Abnormal Platelet Reactivity Is a Coronary Artery Disease Risk Equivalent

Since both, High- and Low-platelet reactivity were associated with increased mortality, we next pooled these patients and compared mortality risk with the risk of patients with angiographically verified coronary artery disease (CAD). We stratified patients into four groups based on the presence and absence of CAD and High/Low platelet reactivity, respectively, and calculated survival curves. Patients with CAD and High/Low platelet reactivity had the highest mortality risk while patients without CAD and normal platelet reactivity had the lowest risk. (Figure 2A,B) Strikingly, patients with CAD and normal platelet reactivity exhibited a similar cardiovascular and all-cause mortality risk compared to patients without CAD but High/Low platelet reactivity, suggesting that abnormal platelet reactivity is a CAD risk equivalent in our study population (log-rank test: *p* = 0.86 for cardiovascular mortality [*p* = 0.16 for all-cause mortality]. This risk-equivalence persisted when CAD mortality was compared with High- and Low-platelet reactivity separately (Appendix A). 

### 3.4. Relative Importance of Risk Markers in Patients with High- and Low-Platelet Reactivity

To investigate potential differences in risk profiles among patients with High- and Low-platelet reactivity, we analyzed the association of predefined risk markers in a Cox model for all groups separately (Figure 3A,B). Given the risk-equivalence of High- and Low-platelet reactivity to CAD, we focused on risk markers considered to be of importance in cardiovascular disease prevention [17]. Age and Charlson Comorbidity Index (CCI) were among the strongest relative contributors in the Cox model irrespective of platelet reactivity status. In addition, eGFR and HbA1c were relatively high-ranked contributors in all groups (all R^2^ > 0.01). However, HbA1c ranked highest in patients with High-platelet reactivity compared to Low-platelet reactivity and the reference group in particular with respect to cardiovascular mortality (High platelet reactivity: R^2^ 0.037, Low platelet reactivity: R^2^ 0.015, Reference Group: R^2^ 0.011). In addition, high-sensitive CRP and antiplatelet therapy with Aspirin was associated with cardiovascular mortality in the high platelet reactivity group but did not reach the threshold of R^2^ ≥ 0.01 in the remaining groups (High platelet reactivity: R^2^ 0.015 [hsCRP]; 0.015 [Aspirin], Low platelet reactivity: R^2^ 0.003 [hsCRP]; 0.001 [Aspirin], Reference Group: R^2^ 0.008 [hsCRP]; 0.003 [Aspirin]). Other cardiovascular risk factors, including smoking and LDL-C, were of relatively little importance among all groups, irrespective of cardiovascular and all-cause mortality (all < R^2^ 0.01).

### 3.5. Risk Assessment in Patients with Abnormal Platelet Reactivity

Next, we selected the four highest ranked risk-modifiers (i.e., eGFR, HbA1c in patients with diabetes, hsCRP, and Aspirin therapy), identified by the relative weight analysis and tested in a hypothesis-generating approach whether these may modulate elevated cardiovascular or overall mortality risk in patients with High and Low platelet reactivity (Figure 4A,B). Patients with either High- or Low-platelet reactivity were stratified based on the presence or absence of these modifiers. Mortality risk was compared to the risk of the reference group in the absence of a risk modifier or the presence of therapy (i.e., the group with the lowest mortality risk). Following this strategy, we found that an eGFR > 60 mL/min, HbA1c < 7.0% (in subjects with diabetes mellitus), and hsCRP < 3 mg/L were associated with a cardiovascular and all-cause mortality risk reduction in patients with either High- or Low-platelet reactivity (Figure 4A,B). Interaction analysis indicated that the effect of risk modification was equal for all groups (all *p* > 0.05). In contrast, interaction analysis of Aspirin therapy indicated a higher treatment effect in patients with High-platelet reactivity compared to the reference group (*p* for interaction: 0.02 [all-cause mortality: <0.001] that was not present in patients with Low-platelet reactivity (*p* for interaction: 0.83 [all-cause mortality: 0.20]. Prediction functions for eGFR, HbA1c, and hsCRP to assess the mortality risk modulation on a linear scale are presented in the Appendix A.

## 4. Discussion

The present explorative study demonstrates that High- and Low-ADP-induced platelet reactivity measured by CD63 expression, are predictors of cardiovascular and all-cause mortality in the LURIC study and risk-equivalent to the presence of CAD. In a hypothesis-generating approach, we explored potential modifiers of elevated cardiovascular and overall mortality risk in patients with High- and Low- platelet reactivity and identified (i) normal-near glucose control in subjects with diabetes mellitus, (ii) preserved kidney function, and (iii) low hsCRP levels as potential risk modifiers. The presence of these risk-modifiers was associated with a cardiovascular risk reduction in all patients, however irrespective of platelet reactivity status. In contrast, patients with High-platelet reactivity had a significantly higher cardiovascular and all-cause mortality risk reduction when treated with Aspirin. 

High- and Low-platelet reactivities predict atherothrombosis, bleeding, and mortality risk under various conditions assessed by multiple assays, including the VerifyNow, Multiplate analyzer, Light transmission aggregometry, and the VASP assay [1,2,3,4,5,9,10,11,19,20,21,22,23,24]. While most of the present studies investigated the role of platelet reactivity to assess drug resistance in highly-selected populations on dual antiplatelet therapy, the LURIC study encompasses an unselected, P2Y_12_ naïve population, scheduled for angiography in a real-world setting. Given the long-term follow-up of 8.4 years, our approach allowed us to investigate the prognostic role of platelet reactivity in a population with a high number of fatal endpoints (29.3% mortality events; 18.8% of cardiovascular cause). Remarkably, we found that both High- and Low-platelet reactivity were equally associated with an increased risk, as platelet reactivity demonstrated a U-shaped relationship with cardiovascular and all-cause mortality. Therefore, our study extends previous observations by Puurunen et al. and Thaulow et al., who focused on the prognostic role of High-platelet reactivity in comparable cohort studies [2,5]. We found all three activity markers, including Fibrinogen-binding, CD62P, and CD63 to be associated with an increased mortality risk. However, CD63 demonstrated better discrimination for patients with low-level platelet reactivity and was therefore used for all analyses presented. Of note, Fibrinogen-binding is an indirect marker of the active conformation of αIIbβIII; CD62P indicates alpha granules release [25] while CD63 indicates dense granules release and therefore, all markers may represent different phenotypes of platelet activation [26]. Fibrinogen-binding is a relatively imprecise marker of active αIIbββIII and may therefore diminish the ability of the marker to detect platelet with low-platelet reactivity. In addition, rapid CD62P shedding [27,28] may explain the inferior ability of CD62P to identify patients with low platelet reactivity, since patients with increased shedding and true low-platelet reactivity will both have decreased levels of CD62P. Therefore, some studies suggest that CD63 expression may be a better marker for ongoing platelet activation which may have a different prognostic meaning [29]. However, the comparison of Fibrinogen-binding, CD62P, and CD63 was beyond the scope of this study and remains speculative. 

Impaired platelet reactivity is associated with increased bleeding risk in numerous studies [30,31,32,33]. While fatal bleeding events were not present in the studied population, non-fatal bleeding events were not explicitly recorded in the LURIC study. Indeed, the ASCEND trial of aspirin for primary prevention in patients with diabetes mellitus demonstrated that patients with the highest risk of non-fatal bleeding events were those with the highest cardiovascular mortality risk [34]. However, the reason for the increased rate of cardiovascular mortality in patients with Low-platelet reactivity is incompletely understood. It can be speculated that patients with Low-platelet reactivity have an increased rate of intramural hematomas of the coronary vessels that increases their cardiovascular event risk. In addition, if Low-platelet reactivity is an indicator of non-fatal bleeding, anti-platelet therapy may have been stopped prematurely in these patients, leaving them at increased cardiovascular event risk. Considering the available literature, it needs to be underlined that these patients represent a yet incompletely understood patient group that is clinically challenging and deserves further attention in future trials. 

Most importantly, patients with High- and Low-platelet reactivity had the same cardiovascular and all-cause mortality risk as patients with CAD and normal platelet reactivity, suggesting that abnormal platelet reactivity is a CAD risk equivalent. In addition, the analogous progression of the survival curves over time underlines the prognostic importance of platelet reactivity as a biomarker. However, in contrast to established treatment algorithms in patients with CAD, therapeutic strategies for patients with abnormal platelet reactivity are poorly defined. Given the uncertainty of therapeutic strategies, we set out to explore whether certain risk markers or therapies may modulate cardiovascular or overall mortality risk in patients with abnormal platelet reactivity. To this end, we employed a combined strategy of a relative weight analysis, stratified cox proportional hazard models, and prediction functions by restricted cubic splines to identify potential risk factors and markers, inspired by a similar strategy by Rawshani and colleagues [16]. For stratification and evaluation of risk groups, we used treatment recommendations from the ESC for cardiovascular disease prevention and observational studies [17,18]. Following this strategy, we identified the presence of antiplatelet therapy by Aspirin as a potential risk modifier that was associated with a higher risk reduction in patients with High-platelet reactivity compared to patients in the Reference- and Low-platelet reactivity groups. This observation is strengthened by the fact that Aspirin treatment rates were similar among all platelet reactivity risk groups. However, we cannot exclude that Aspirin therapy constitutes a confounder in the LURIC study for pre-identified high-risk patients that may have received better treatment. Therefore, it remains unclear whether the risk reduction is a direct treatment effect or indicates better medical treatment and thus should be interpreted with caution. However, given the ongoing debate on Aspirin therapy in primary prevention [34,35,36], it is attractive to speculate that ADP-induced CD63 expression may constitute a potential marker to identify patients that may benefit more from Aspirin therapy. Nevertheless, prospective trials are needed to address this question. 

In addition, we identified (i) HbA1c < 7.0% in patients with diabetes mellitus, (ii) eGFR > 60 mL/min/1.73 m^2^, and (iii) hsCRP < 3mg/L as potential risk modifiers. The presence of these risk modifiers was associated with a risk reduction that was similar in all patients and therefore indicated effects independent of platelet reactivity. However, given the increased absolute mortality risk of patients with either High- or Low-platelet reactivity, the net risk reduction is higher in patients with abnormal platelet reactivity. Of note, given the nature of this study, we were unable to demonstrate causality, and prospective trials are needed to validate these findings. 

Our study has some limitations: We included a Caucasian patient population undergoing coronary angiography without any major non-cardiac diseases. Therefore, our data cannot be generalized to populations of other ethnicities, younger ages, or those with additional comorbidities. In addition, the LURIC study only recorded fatal outcomes. Therefore, non-fatal outcomes could not be evaluated (i.e., myocardial infarction, stent-thrombosis, bleeding, etc.). In addition, our study included more male subjects (68%) than females, and therefore a bias toward the male sex cannot be excluded. In this study, we utilized the degranulation markers CD62P and CD63 as markers of ADP-induced platelet reactivity. Despite the widely used platelet activity markers in experimental research, these markers are less well characterized in a clinical setting. Nevertheless, we were able to demonstrate that (i) both markers increased expression in response to stimulation (Appendix A) and (ii) both markers identified patients at risk of cardiovascular and all-cause mortality. 

In summary, our study extended current knowledge on the prognostic importance of High- and Low-platelet reactivity and suggests that both are CAD risk equivalents. In addition, we demonstrated a close association of mortality in patients with High and Low platelet reactivity with renal function, glucose control, inflammation. Our data indicated that patients with High-platelet reactivity may benefit more from anti-platelet by Aspirin therapy. 

## Figures and Tables

**Figure 1 jcm-12-01913-f001:**
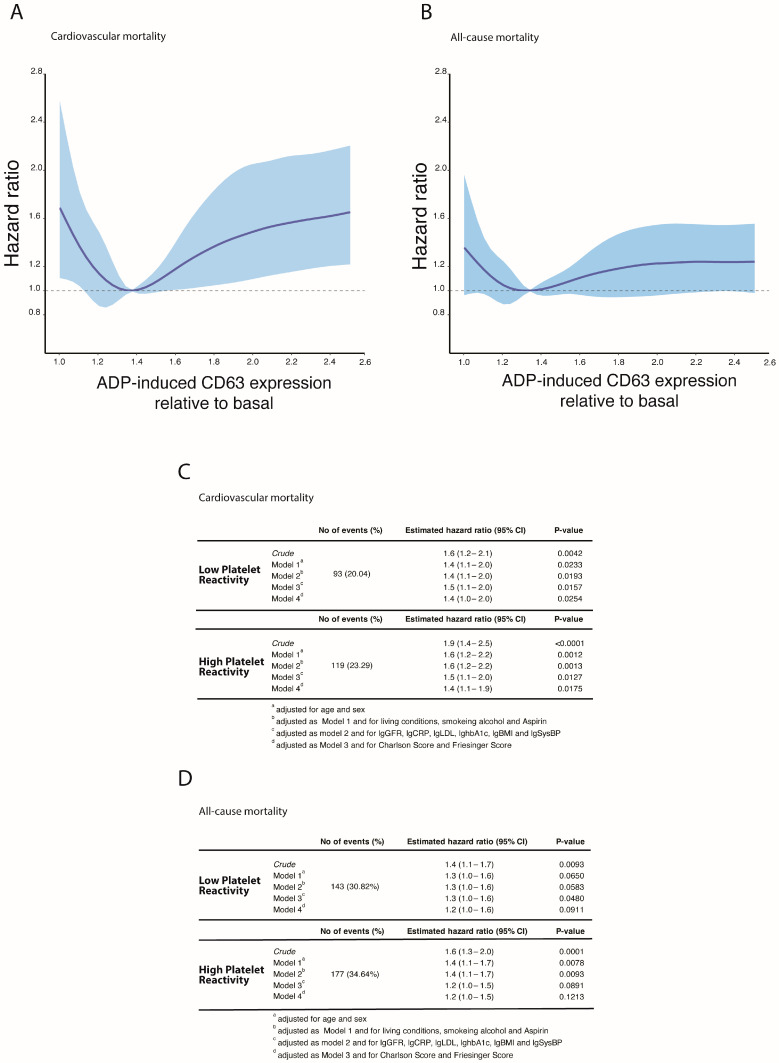
Platelet reactivity and cardiovascular and all-cause mortality in the LURIC study. Association between ADP-induced CD63 expression and cardiovascular and all-cause mortality in 1520 patients. ADP-induced CD63-expression was modeled with restricted cubic splines for the hazard of cardiovascular (**A**) and all-cause mortality (**B**). Dark line represents the hazard function while the shaded area denotes the 95% confidence interval. All models presented were adjusted for age, sex, living conditions (marital status), alcohol intake, smoking status, anti-platelet therapy, lg-GFR, lg-CRP, lg-systolic lg-bloodpressure, Charlson Comorbidity Index and Friesinger Score (model 4). (**C**,**D**) Crude and stepwise risk-factor adjusted hazard ratios of cardiovascular (**C**) and all-cause mortality (**D**) according to Low- and High-platelet reactivity.

**Figure 2 jcm-12-01913-f002:**
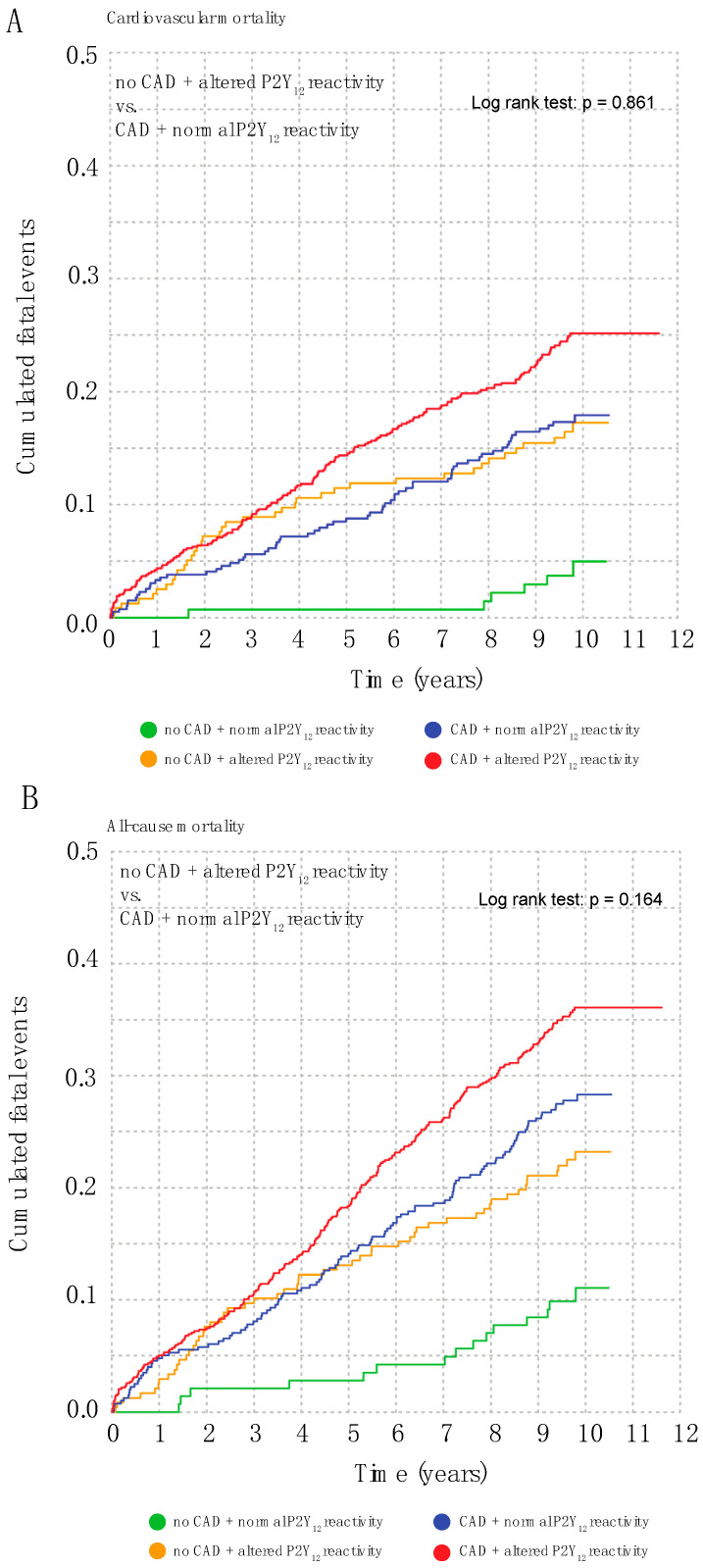
High- and Low-platelet reactivity is risk-equivalent to coronary artery disease. Kaplan-Meier cumulative event curves in 1520 patients for cardiovascular (**A**) and all-cause mortality (**B**) in patients stratified by the presence (i.e., at least ≥1 angiographically verified stenosis of ≥20%) and absence of coronary artery disease and High/Low platelet reactivity.

**Figure 3 jcm-12-01913-f003:**
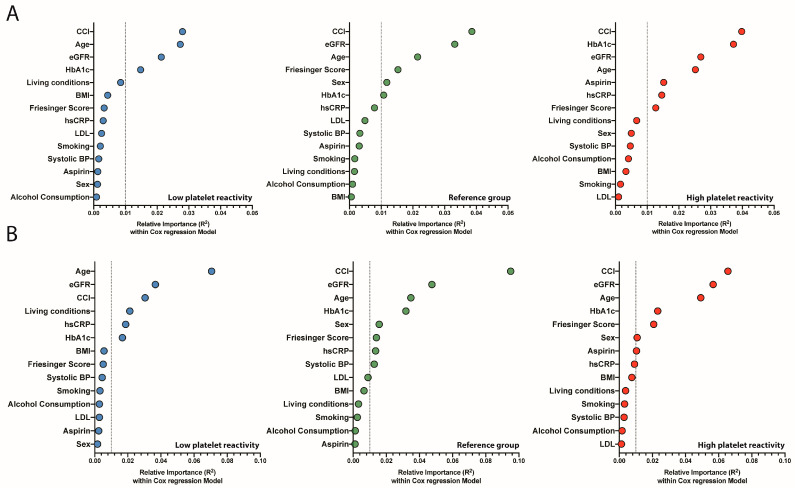
Relative Importance of risk factors that predict cardiovascular and all-cause mortality. Relative Importance analysis for 464 patients in the Low platelet reactivity group, 545 in the Reference group, and 511 in the High platelet reactivity group. The estimated relative importance denotes the strength of the association (R^2^) for risk-factor variables for predicting cardiovascular (**A**) and all-cause mortality (**B**) among patients with High- and Low-platelet reactivity and the Reference group. The Cox model applied for this relative weight analysis is similar to model 4 in Figure 1. BMI—body mass index, eGFR—glomerular filtration rate, LDL-C—Low-density Lipoprotein, hsCRP—high-sensitive C-reactive Protein, CCI—Charlson Comorbidity Index; Blue circle—Low platelet reactivity; Green circle—Reference Group; Red circle—High platelet reactivity.

**Figure 4 jcm-12-01913-f004:**
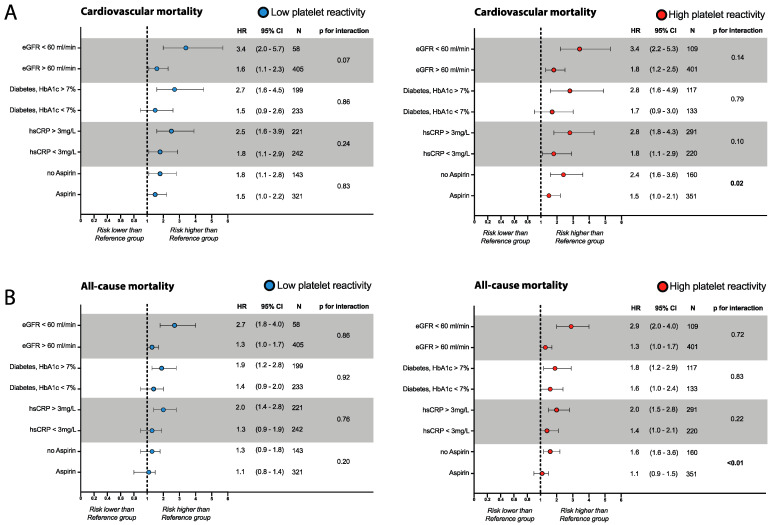
Risk factor-stratified hazard ratios for cardiovascular and all-cause mortality among patients with altered platelet reactivity and the reference group. Forest plot for cardiovascular (**A**) and all-cause mortality (**B**) based on key risk factors/therapies identified by relative weight analysis. Patients with High-(red dot, 511 patients) and Low-(blue dot, 464 patients) platelet reactivity were stratified according to the presence or absence of the appropriate factor. Risk was compared to the risk of the Reference group (545 patients). Continuous variables were a priori categorized based on available guidelines or previous studies (i.e., hs-CRP). The cardiovascular mortality risk in each group was compared to the reference group in the absence of the factor of interest/presence of the therapy of interest. Models were adjusted for age and sex similar to model 1 in Figure 1.

**Table 1 jcm-12-01913-t001:** Patient characteristics stratified by platelet reactivity.

Characteristics	All Patients	Low P2Y12 Reactivity	Reference Group	High P2Y12 Reactivity	*p* Value
	(n = 1520)	(n = 464)	(n = 545)	(n = 511)	
ADP-induced CD63 expression (relative to basal)		<1.22	1.22–1.50	>1.50	
Age (Years)	64.05 (56.92–71.02)	63.65 (57.15–71.05)	63.5 (55.28–69.98)	65.79 (58.32–71.91)	**0.0001**
Sex (male)	1037 (68.22%)	325 (70.04%)	382 (70.09%)	330 (64.58%)	0.0975
Body Mass Index	27.2 (24.87–29.88)	27.02 (25.09–29.96)	27.26 (24.74–29.57)	27.4 (24.71–30.41)	0.6664
**Cardiovascular risk factors**					
Hypertension	868 (57.11%)	256 (55.17%)	311 (57.06%)	301 (58.9%)	0.4993
Hypercholesterolemia	1043 (68.62%)	316 (68.1%)	372 (68.26%)	355 (69.47%)	0.8767
Diabetes mellitus	648 (42.63%)	182 (39.22%)	215 (39.45%)	251 (49.12%)	0.0014
Chronic kidney disease	223 (15.33%)	57 (12.72%)	61 (11.75%)	105 (21.52%)	**<0.0001**
Family History of premature MI	163 (10.72%)	59 (12.72%)	63 (11.56%)	41 (8.02%)	**0.0405**
Premature MI	254 (16.71%)	71 (15.3%)	106 (19.45%)	77 (15.07%)	0.1065
Alcohol intake	912 (60%)	300 (64.66%)	315 (57.8%)	297 (58.12%)	**0.0479**
Smoking	340(22.1%)	110(23.7%)	118(21.7%)	109(21.3%)	0.6290
**Scores**					
Friesinger Score	5 (2–8)	5 (2–8)	4 (1–8)	5 (2–8)	0.1994
Framingham Score	15 (13–17)	15 (13–16)	15 (12–16)	15 (14–17)	**0.0002**
Marschner Score	0.14 (0.1–0.2)	0.14 (0.09–0.2)	0.15 (0.1–0.21)	0.14 (0.1–0.19)	0.2498
Charlson Score	1 (1–2)	1 (1–2)	1 (1–2)	2 (1–3)	**0.0002**
**Death**					
All-cause death	452 (29.74%)	143 (30.82%)	132 (24.22%)	177 (34.64%)	**0.0008**
Cardiovascular death	284 (18.68%)	93 (20.04%)	72 (13.21%)	119 (23.29%)	**0.0001**
**Medical History**					
Acute Coronary Syndrom at baseline	410 (27.61%)	128 (28.38%)	139 (26.18%)	143 (28.43%)	0.6560
Coronary artery disease	1141 (75.07%)	347 (74.78%)	403 (73.94%)	391 (76.52%)	0.6222
Coronary artery bypass graft	184 (12.11%)	50 (10.78%)	66 (12.11%)	68 (13.31%)	0.4761
History of PTCA	335 (22.04%)	110 (23.71%)	114 (20.92%)	111 (21.72%)	0.5526
Peripheral artery disease	114 (7.5%)	37 (7.97%)	36 (6.61%)	41 (8.02%)	0.6152
Stroke	120 (7.89%)	35 (7.54%)	37 (6.79%)	48 (9.39%)	0.2819
Heartfailure	500(32.6%)	158(34.1%)	178(32.7%)	160(31.3%)	0.6590
HFpEF	264(17.2%)	74/15.9%)	78(14.3%)	82(16.0%)	0.6750
HFrEF	236(15.4%)	84(18.1%)	100(18.3%)	78(15.3%)	0.3430
**Laboratory parameters**					
Platelets (nL)	225 (186–270)	223 (184–266)	227 (189–268)	226 (185–273.5)	0.6755
Mean platelet volume (fl)	8.9 (8.2–9.6)	9 (8.2–9.7)	8.9 (8.2–9.6)	8.8 (8.25–9.6)	0.3991
High sensitive CRP (mg/dL)	3.08 (1.29–7.82)	2.86 (1.24–6.79)	2.7 (1.18–7.58)	3.83 (1.63–9.04)	**0.0002**
Glycosylated hemoglobin (%)	6.2 (5.8–6.8)	6.1 (5.75–6.7)	6.2 (5.8–6.7)	6.3 (5.8–6.97)	0.0116
Glomerular Filtration Rate (mL/min/1.73m^2^)	81.97 (67.64–94.75)	83.15 (69.21–97.03)	83.61 (70.57–97.18)	78.26 (61.96–89.5)	**<0.0001**
Triglycerides (mg/dL)	145 (109–200.5)	150 (112–209)	143 (108–196)	143 (110.5–198.5)	0.1380
LDL Cholesterol (mg/dL)	112 (90–135)	112 (93–136)	113 (92–138)	109 (88–132)	0.0865
HDL Cholesterol (mg/dL)	38 (31–45)	38 (31–45)	38 (32–45)	38 (31–45)	0.7093

LPR vs. Reference group *p* < 0.05; HPR vs. Reference group *p* < 0.05; LPR vs. HPR *p* < 0.05. Patient characteristics of 1520 patients included in the LURIC Study stratified by Low-, Reference- and High-platelet reactivity. Continuous variables are expressed as mean ± SD or median (Q1–Q3) in case of skewed data. Categorical variables are shown as absolute and relative frequencies. LPR—Low platelet reactivity and HPR—High platelet reactivity. Low platelet reactivity vs. Reference group tested; High platelet reactivity vs. Reference group tested; Low platelet reactivity vs. High platelet reactivity tested. Bold *p* values indicate *p* < 0.05.

**Table 2 jcm-12-01913-t002:** Baseline medication stratified by platelet reactivity.

Baseline Medication	All Patients	Low P2Y12 Reactivity	Reference Group	High P2Y12 Reactivity	*p* Value
	(n = 1520)	(n = 464)	(n = 545)	(n = 511)	
Aspirin	1046 (68.82%)	321 (69.18%)	374 (68.62%)	351 (68.69%)	0.9799
Phenprocoumon/Warfarin	108 (7.11%)	37 (7.97%)	36 (6.61%)	35 (6.85%)	0.6794
Statin	727 (47.83%)	221 (47.63%)	259 (47.52%)	247 (48.34%)	0.9635
Calcium Channel blocker	77 (5.07%)	13 (2.8%)	25 (4.59%)	39 (7.63%)	**0.0024**
ACEi/ARB	809 (53.22%)	236 (50.86%)	290 (53.21%)	283 (55.38%)	0.3724
Diuretic	454 (29.87%)	129 (27.8%)	148 (27.16%)	177 (34.64%)	**0.0159**
Beta Blocker	934 (61.45%)	286 (61.64%)	332 (60.92%)	316 (61.84%)	0.9572

Baseline medication of 1520 patients included in the LURIC Study stratified by Low-, Reference- and High-platelet reactivity. Data are shown as absolute and relative frequencies. ACEi—ACE inhibitor and ARB—angiotensin II receptor blocker. LPR—Low platelet reactivity and HPR—High platelet reactivity. High platelet reactivity vs. Reference group tested; Low platelet reactivity vs. High platelet reactivity tested. Bold *p* values indicate *p* < 0.05.

## Data Availability

Data will be provided upon reasonable request.

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
