# Peer review of "Platelet Reactivity and Cardiovascular Mortality Risk in the LURIC Study"

_jcm, 2023, doi:10.3390/jcm12051913_

Round 1

Reviewer 1 Report

Berger et al. applied a large cohort sample (with CAD profile at baseline) and ADP and TRAP6 platelet reactivity (PR) measures by flow cytometry to examine associations with future CVD and total mortality. This is somewhat related to their prior work which stratified risk based on TMAO levels and PR though they adopted a somewhat different statistical approach and focused on CD63 more so than CD62P here. A strength of the study is the relatively large sample size, prospective follow-up and measurement of possible confounders and other risk factors.

Major

1)    There are some statistical concerns with the work. The authors apply a somewhat atypical approach to risk estimation, deriving low and high thresholds from cubic splines that already examine HR across the spectrum of samples (Figure S3). They then discard the CD62P from consideration based on lesser discrimination. However, in a sense they have already analysed this data. From a statistical standpoint they have in essence examined 8 conditions (2 agonists X 2 markers X 2 outcomes). The confidence intervals on the Low and High group associations are quite wide and the P-values of marginal significance, being p=0.025 and p=0.018, respectively. If one accounts for multiple testing these associations in essence would become non-significant, thus this tempers the overall enthusiasm and impact of the work.

2)    The authors have discarded ~9% of the sample data with activation levels <1.0%. The conclusion is this is done largely because of concerns of higher pre-activation (basal levels as shown). It could also be a result of possible sample preparation issues. However, there are possible concerns with this approach. It is likely you are discarding some samples that truly have low activation values and therefore you are biasing or losing power in your Low reactivity group. Likewise, if this assay were to be used in a clinical or other population setting would others be expected to lose 9% of samples that they can do risk estimation on? This seems a concerningly large proportion.

3)    The authors previously reported an association of higher CD62P PR with CVD/total mortality (Berger et al.) whether stratified on TMAO or not. In that work they used tertiles rather than cubic spline models. Here CD62P is not fully analyzed and presented. It seems relevant whether with a different approach those results remain significant or not.

4)    There is not discussion made of the biology of CD62P vs. CD63 as markers of function. CD62P has traditionally been more widely applied in past studies (as a marker of alpha granule exocytosis). CD63 is less often applied and thought of as a dense granule marker (the dense granules containing most of the platelet cellular ADP that is released and contributes to secondary activation). I think you need to discuss these distinctions and speculate on the meaning of the differences you observed.

5)    Was follow-up surveillance for non-fatal CVD events done as well? It seems a relevant question to ask whether CD62P and/or CD63 display similar or different associations with non-fatal (or non-fatal + fatal) CVD outcomes since they also carry high population prevalence and clinical burdens.

6)    There are a number of concerns with flow cytometry approach and methods (at least from information provided). This is of key importance because this is the core focus and biomarker of the paper. The current presentation does not meet standards for minimal presentation of information with Flow Cytometry experiments. These concerns are addressed in subpoints here:

a.     Samples were stimulated with ADP or TRAP for 5 mins, was this adequate to induce both alpha and lysosome release? Please provide supporting supplementary data.

b.     Please include details of the monoclonal antibody isotypes, clones, fluorochrome conjugates (was CD41 also conjugated with FITC), and final titrated concentrations used for all antibodies used.

c.     Please also provide a representative flow cytograph illustrating the gating of CD41 positive platelets following agonists stimulation.

d.     In your original 2001 LURIC cohort design paper you describe your measurements. From this it is apparent that you measured more markers including CD42, CD61 and fibrinogen binding (by PAC1?) but these have not been included in your analyses which is somewhat puzzling. Are there issues with that data? Some people prefer CD61 as a marker since CD41/42 epitopes can be internalized into platelets (depends also on which antibody clones/binding sites were used).

e.     In the 2001 paper the Coulter EPICS XL-MCL is given as the instrument. This is a different instrument from the one described here (Coulter FC 500). Were several instruments used or is there an error somewhere? If several instruments were used were there quality control metrics/standardization measures taken?

f.      The concentration of the isotypic control antibodies used should be provided, and representative flow cytographs illustrating the gating strategy and the use of the mentioned isotypic controls could be included. This is in keeping for Minimal standards established for flow cytometry experiments.

g.     Please provide details on the blood collection (1st tube drawn, middle or last draw?)

Minor

1)    One limitation of the current study and the prior trials is sex-bias toward males. Since platelet count (and function) is thought to have some significant differences between men and women this seems to be an important difference that merits some discussion and future consideration.

2)    The Abstract could use some re-writing. Particularly, the last few sentences. You should indicate directionality (e.g., lower inflammation, improved kidney function). The last sentence essentially repeats what is stated 3 sentences before. There is also an assumption of causality in some of the statements – e.g., “aspirin was most effective”. These results are associations rather than RCTs so it is hard to conclude this was solely due to differing drug efficacy.

3)    In regards to the statements on aspirin efficacy, the efficacy of aspirin in CVD prevention has come under continued question, for example from results of the ASPREE Clinical trial. It is worth considering the balance of studies on aspirin efficacy and how important markers such as CD63 may be, and whether/if they could be part of a picture to stratify efficacy or not and what would be required to demonstrate that.

4)    The introduction describes this as a “pilot study”. Is that really accurate since the data was collected many years ago in a relatively large sample? It seems more like a relatively large completed cohort study, unless you plan to launch a larger study based on this that you have not specifically described.

5)    Figure S4 appears to have an error in that the lower right plot should be labeled All-Cause Mortality rather than CVD Mortality

6)    TRAP should be TRAP-6 I think

7)    Please give vendors for all items in Methods including platelet agonists

Author Response

We would like to thank the reviewer for their thorough evaluation of our manuscript. Please find attached our detailed response

Reviewer 2 Report

In a P2Y12-inhibitor naïve patient population referred for coronary angiography, the authors found that ADP-induced platelet reactivity, as reflected by the expression of dense granule glycoprotein CD63, was a strong predictor of cardiovascular and all-cause mortality. Interestingly, both the low and high platelet reactivities were related with high mortality. This is an interesting observation, expanding previous knowledge of high-platelet reactivity being associated with higher mortality.

Major comments:

Should describe more detail of blood collection, venous or arterial, type of anticoagulant tubes used, whether the first tube of blood was discarded, etc. Platelet can be easily activated when an improper blood collection method is used.

Was there any measure taken trying to normalize the fluorescence intensity ratio in the platelet reactivity test? The between-day variation of the assay could be troublesome for result interpretation since this study lasts several years. Is there data on the inter-assay variability of the assay?

Minor:

Page 2, “Study design, participants and clinical characterization” when describing the definition of Clinically relevant coronary artery disease (CAD), it was written that CAD was defined as the occurrence of ≥1 stenosis of ≥50% in ≥1 of 15 coronarys egments. I believe it should be 20%, as described in the supplement.

Figure 1, living condition marital status, please keep consistency.

Labels A and B are missing in figure 3 and 4.

Author Response

We would like to thank the reviewer for their thorough evaluation of our manuscript. Please find a detailed response in the attached file. 

Reviewer 3 Report

In this manuscript by Berger et al. entitled “Platelet reactivity and cardiovascular mortality risk in the LURIC study”, the authors aim to determine the association of platelet reactivity and cardiovascular mortality risk. For this purpose, the authors measured ADP-induced CD62P and CD63 expression on platelets using flow-cytometry in 1520 patients who were referred for coronary angiography in the Ludwigshafen Risk and Cardiovascular Health Study (LURIC). The authors found that both high- and low-platelet reactivity were associated with all-cause and cardiovascular mortality. 

Comments to the authors:

·      The methods section on platelet reactivity testing should provide more details. The authors chose a test of ex vivo platelet activation that is susceptible to preanalytical variation such as type of blood collection tube, centrifugation speed, sample processing time etc. The authors should also explicitly state what they mean by “relative to basal”, although it can be assumed that they mean CD63 on platelet membrane expression post agonist stimulation relative to CD63 on platelet membrane expression before agonist stimulation.

·      The description of how patients were stratified into high- and low-platelet reactivity groups is confusing and can be made clearer. It seems the authors chose the stratification based on the U-shaped relationship of CD63 expression with mortality hazard ratios, using CD63 levels below (low reactivity) or above the cut-off (high reactivity) where hazard ratios are significantly deviating from 1. The authors should state in the manuscript text what the precise levels of CD63 were that they used for stratification (based on table 1, this appears to be <1.22; 1.22-1.50; >1.50).

·      The authors merged non-aspirin users (31%) and aspirin users (69%) into the same analysis. Aspirin use is a major determinant of the degree of platelet activation even when ADP is used as an agonist. Although the groups of low reactivity, high reactivity and the reference patients appear to contain a similar number of aspirin users (which is surprising), it is unclear to which extent aspirin therapy confounds the presented analysis. It may therefore be interesting to add an analysis that looks at aspirin users and non-aspirin users separately.

·      Strikingly, patients with CAD and normal platelet reactivity exhibited a similar cardiovascular and all-cause mortality risk compared to patients without CAD but High/Low platelet reactivity, suggesting that abnormal platelet reactivity is a CAD risk equivalent in our study population (log rank test: p=0.84 for cardiovascular mortality [p=0.13 for all-cause mortality]).” Although the Kaplan Meier curves (Fig. 2) appear to confirm the trend described by the authors, the corresponding p-values are not significant. Could you please clarify this.

·      The authors provide no explanation for the surprising finding that not only high but also low platelet reactivity is associated with increased mortality? The manuscript would benefit if the discussion on this finding could be expanded.

Author Response

We would like to thank the author for their thorough evaluation of our manuscript. Please find attached a file with a detailed response to all Reviewer comments. 
